# Removal of false positives in metagenomics-based taxonomy profiling via targeting Type IIB restriction sites

Zheng Sun [1,6], Jiang Liu[2,6], Meng Zhang [3,6], Tong Wang[1], Shi Huang[4], Scott T. Weiss[1] & Yang-Yu Liu [1,5] ✉

Accurate species identification and abundance estimation are critical for the interpretation of whole metagenome sequencing (WMS) data. Yet, existing metagenomic profilers suffer from false-positive identifications, which can account for more than 90% of total identified species. Here, by leveraging species-specific Type IIB restriction endonuclease digestion sites as reference instead of universal markers or whole microbial genomes, we present a metagenomic profiler, MAP2B (MetAgenomic Profiler based on type IIB restriction sites), to resolve those issues. We first illustrate the pitfalls of using relative abundance as the only feature in determining false positives. We then propose a feature set to distinguish false positives from true positives, and using simulated metagenomes from CAMI2, we establish a false-positive recognition model. By benchmarking the performance in metagenomic profiling using a simulation dataset with varying sequencing depth and species richness, we illustrate the superior performance of MAP2B over existing metagenomic profilers in species identification. We further test the performance of MAP2B using real WMS data from an ATCC mock community, confirming its superior precision against sequencing depth. Finally, by leveraging WMS data from an IBD cohort, we demonstrate the taxonomic features generated by MAP2B can better discriminate IBD and predict metabolomic profiles.

During the past decades, advances in metagenomics have dramatically increased our understanding of microbial life and greatly promoted developments related to food production, agriculture, environmental remediation, drug discovery and human health[1]. Currently, culture-independent high-throughput sequencing (e.g., amplicon sequencing and whole metagenome sequencing) is the predominant technique for metagenomics and has played a pivotal role in identifying causes of antibiotic resistance[2], infectious disease outbreaks[3], and cancer

oncogenesis[4,5]. It is well known that amplicon sequencing suffers from off-target amplification[6], biased abundance estimation, limited taxonomic resolution, insensitivity to degraded DNA, and an inability to simultaneously capture all microorganisms (e.g., bacteria, fungi, archaea, and virus) in one sequencing[7]. Notably, whole metagenome sequencing (WMS) can capture all microorganisms at the species (or even strain) resolution, hence having greater potential for clinical practice than amplicon sequencing. However, false-positive

[1]Channing Division of Network Medicine, Department of Medicine, Brigham and Women's Hospital and Harvard Medical School, Boston, MA, USA. [2]Qingdao OE Biotechnology Company Limited, Qingdao, Shandong, China. [3]Key Laboratory of Dairy Biotechnology and Engineering, Ministry of Education, Inner Mongolia Agricultural University, Hohhot, China. [4]Faculty of Dentistry, The University of Hong Kong, Hong Kong, China. [5]Center for Artificial Intelligence and Modeling, The Carl R. Woese Institute for Genomic Biology, University of Illinois at Urbana-Champaign, Champaign, IL, USA. [6]These authors contributed equally: Zheng Sun, Jiang Liu, Meng Zhang. ✉e-mail: yyl@channing.harvard.edu

identification presents a major challenge for the interpretation of WMS data[8].

The false-positive identification issue in WMS data could be influenced by both experimental and computational factors. For example, contamination of the samples can be introduced from laboratory kits, reagents, or the environment during sample collection, DNA extraction, handling, storage, or sequencing, which can yield high numbers of spurious identifications[9-11]. The false positives due to contaminations in the wet-lab environment could be largely avoided by using data from multiple control groups as filters[12]. However, computational methods (e.g., reference-based metagenomic profilers, which attempt to efficiently decode WMS reads without assembly) were found to have a more significant effect size on the false-positive identification issue in WMS data. For example, a similar number of false positives was identified by comparing simulated WMS data and real WMS data of an ATCC mock community[13]. No state-of-the-art metagenomic profilers excelled in taxon identification and abundance estimation at the species level[14]. Such bottleneck faced by traditional metagenomic profilers is due to their reliance on universal single-copy markers or whole microbial genomes as references. This often results in challenges like missing markers or multi-alignment of short reads. In contrast, we found that species-specific Type IIB restriction endonuclease digestion sites, which are evenly and abundantly distributed across microbial genomes, outnumber universal markers and can naturally avoid the multi-alignment problem. Thus, we believe they have the potential to serve as effective reference markers to address the above bottleneck.

Here, we present MAP2B (MetAgenomic Profiler based on type IIB restriction site), a metagenomic profiler that can effectively eliminate false positives and hence generate higher precision and more accurate taxonomic profiles from WMS data. In this study, we first illustrate the pitfall of using relative abundances to filter out false positives. To resolve this issue, we propose a more meaningful feature set for determining false positives and establish a false-positive recognition model using simulation data in CAMI2. Then we systematically benchmark the performance of MAP2B in species identification using a series of systematically generated simulation data with varying sequencing depth and species richness based on random microbial genomes in NCBI RefSeq. We then leverage data of an ATCC mock community (MSA 1002) to further validate and demonstrate the precision, accuracy and the potential of MAP2B in dealing with real WMS data. Finally, we demonstrate the power of using MAP2B to better discriminate disease status and predict metabolomic profiles, leveraging WMS data from an IBD cohort[15]. In summary, MAP2B can significantly improve the precision and recall in species identification, which will vastly optimize the decoding of the taxonomic structure in microbiome studies using WMS data, e.g., it will profoundly reduce the false-positive rate and therefore improve the resolution for differential abundance analysis, biomarker detection, phenotype classification, and disease prediction.

## Results

### The pitfall of using relative abundances to filter out false positives

Currently, users only rely on relative abundances generated by existing metagenomic profilers to filter out false positives. However, as shown in Fig. 1a–c, those false positives are not necessarily species of low abundances. Hence, only using relative abundances to filter out false positives will lead to a substantial drop in Precision and Recall. Indeed, the benchmark study of CAMI2 (Critical Assessment of Metagenome Interpretation: second round of challenge)[14] shows that several widely used tools for metagenome analysis, such as Bracken[16], MetaPhlAn2[17], and mOTUs2[18], have an average Precision range of 0.11 to 0.60 and Recall range of 0.62 to 0.67 for three simulated datasets (marine, plant-associated, and strain madness). These results highlight the difficulty

of accurately interpreting metagenomic data, even with state-of-the-art tools. To explicitly demonstrate the issue of using relative abundances to filter out false positives, let's consider the first sample (labeled as No.0) in each of the three CAMI2 simulated datasets. We sorted the identified species based on the descending order of their relative abundances generated by each of the five representative metagenomic profilers with their latest version: MetaPhlAn4[19], mOTUs3[20], Bracken[16], Kraken2[21], and KrakenUniq[22] (see Fig. 1a–c). True and false positives are shown in green and yellow, respectively. False negatives are shown in gray. An ideal profiler should identify all the true positives but nothing else (as shown in the "ground truth" rows in Fig. 1a–c). However, existing metagenomic profilers suffer from false positives and/or false negatives. We clearly see that the highly abundant species are not necessarily the true species, and the false positives are not necessarily species of low abundances. This underscores the pitfall of using only the relative abundance to filter out false positives.

### A feature set in determining false positives

We sought to resolve this false-positive identification issue by selecting a more meaningful set of features to better discriminate true positives from false positives. This feature set includes four features: genome coverage, sequence count, taxonomic count, and G-score, which are defined in order here.

When determining a true positive, reads from present microbes should distribute relatively uniformly across their genomes rather than being concentrated in one or a few genomic regions[22]. Therefore, we hypothesize that the uniformity of genome coverage is a critical metric in determining true positives. It is well known that the endonucleases from the Type IIB restriction-modification systems differ from all other restriction enzymes[23]. In particular, the Type IIB enzymes cleave DNA on both sides of their recognition at fixed positions to cut out the recognition site with iso-length DNA fragments. In a previous study, we demonstrated that Type IIB restriction sites are widely and randomly distributed along microbial genomes[24]. This suggests an efficient method to identify a microbial species and estimate its abundance by profiling the sequence coverage of a fixed set of taxonomic markers of this species in WMS data. First, a true positive should have sufficient reads that can hit the individual markers. Second, the genome coverage (i.e., the sequence coverage of the whole set of species-specific markers in our context) should be as large as possible. Here, we identified ~8607 species-specific 2b tags for each species (i.e., iso-length DNA fragments produced by Type IIB enzyme digestion) based on an integrated genome database of GTDB (Genome Taxonomy Database)[25] and Ensembl Fungi[26] (Supplementary Fig. S1). In this work, we used *CjepI* as a representative type IIB enzyme to perform in silico restriction digestion for each species in GTDB and Ensemble Fungi. For species-$i$ in this integrated database, we denote its total number of 2b tags generated by in silico digestion of its genome as $H_i$. Among the $H_i$ tags, there are $E_i$ tags that are single-copy within species-$i$'s genome and are unique to species-$i$ w.r.t all other species in the database. The genome coverage of species-$i$ (denoted as $C_i$) in a WMS dataset can be formally quantified by the ratio between the number of its observed distinct (or non-redundant) species-specific 2b tags (denoted as $U_i$) in the WMS data and the total number of its species-specific 2b tags (i.e., $E_i$) in the integrated database: $C_i = U_i/E_i$ ("Methods").

As we know, metagenomic profiling often produces two fundamentally different types of relative abundances: sequence abundance and taxonomic abundance. The former describes the proportion of DNA content of a species in a microbial sample, while the latter gives the cell ratio between a species and its entire microbial community[27]. Consider species-$i$ with genome size $L_i$ and ploidy $P_i$ in a microbiome sample. Denote $R_i$ as the DNA content (e.g., the number of metagenomic reads) assigned to species-$i$. The number of cells classified as species-$i$ is simply given by $N_i = R_i/(L_iP_i)$. The sequence and taxonomic

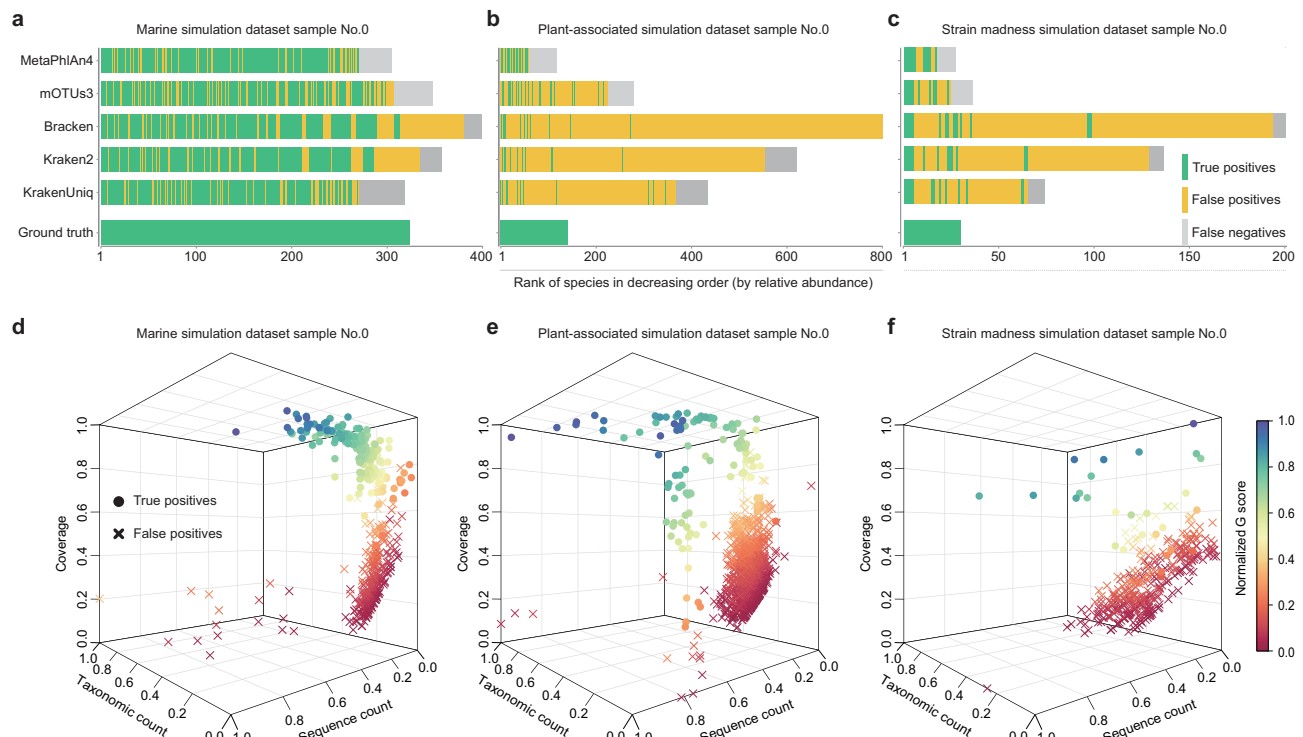

**Fig. 1 | Comparison of the conventional method with MAP2B in false-positive recognition using three CAMI2 simulation data.** To illustrate the pitfall of using relative abundance as the only threshold in species identification and the potential of MAP2B to address this issue, we compared the profiling results generated by different tools. Specifically, the short read WMS data labeled with No.0 in each of the three CAMI2 simulation datasets of **a** marine, **b** plant-associated, and **c** strain madness were processed by state-of-the-art metagenomic profilers (such as MetaPhlAn4, mOTUs3, Bracken, Kraken2, and KrakenUniq) and MAP2B. From left to right in each bar plot, identified species were ranked by their abundance in decreasing order (the x-axis), and the ground truth was also illustrated to demonstrate the pitfall of conventional methods. We again employed the short read WMS data labeled with No.0 in each of the three CAMI2 simulation datasets of **d** marine, **e** plant-associated, and **f** strain madness to illustrate the clear boundary between false positives and true positives using the four features. In the 3D scatter plots, we showed the distributions of true-positive and false-positive species with the x-axis as taxonomic count, the y-axis as sequence count, the z-axis as the coverage of the identified species, and color referring to identified species' G-score. Based on the ground truth, true positives and false positives are shaped as dots and crosses. All four features are scaled by min-max to fit and be visualized in the 3D scatter plots.

abundance of species-$i$ are given by $S_i = R_i / \sum_j R_j$ and $T_i = N_i / \sum_j N_j$, respectively. We have shown that, mathematically, there is no universal or sample-independent algebraic relation between the two types of relative abundances[27]. Hence, they offer different perspectives in describing the relative abundance of a species and would benefit the accuracy of species identification. In this study, we use the sequence count and the taxonomic count as features. Here, the taxonomic count of species-$i$, denoted as $N_i$, is simply calculated as the average count of the sequenced unique 2b tags, i.e., $N_i = Q_i / U_i$, where $Q_i$ is the number of tags unique to species-$i$ in the WMS data. The sequence count of species-$i$, denoted as $R_i$, is defined to be the average count of the inferred unique 2b tags sequenced per read, i.e., $R_i = \tilde{Q}_i / R$, where $\tilde{Q}_i = Q_i / C_i$ is the inferred number of sequenced unique 2b tags in the WMS data, and $R$ is the total number of reads in the WMS data (see "Methods"). Notably, the sequence count $R_i$ (or the taxonomic count $N_i$) shares the flavor of sequence abundance $S_i$ (or the taxonomic abundance $T_i$), but they are not exactly the same.

In our previous study, we have illustrated the G-score of species-$i$, denoted as $G_i$, which is the geometric mean of $Q_i$ and $U_i$, i.e., $G_i = \sqrt{Q_i U_i}$ ("Methods") as an empirically useful feature in determining false positives[24]. In the three-dimensional space spanned by genome coverage, taxonomic count, and sequence count of different species, we can observe a separation between true positives (dot) and false positives (cross) in the three simulated CAMI2 samples (Fig. 1d–f). Moreover, by coloring the identified species with their G-score, we can visually assess the probability of a species being a true positive or false positive. Taken together, these four features provide a promising

foundation to construct a machine-learning classifier to discriminate true positives from false positives.

## The workflow of MAP2B
To eliminate false positives, we developed MAP2B, a metagenomic profiler that takes WMS data as input and generates taxonomic abundances for identified species. Instead of directly estimating the relative abundances of the species through aligning reads against the whole microbial genome or marker genes as existing metagenomic profilers do, we use the following two-round reads alignment strategy.

First, 2b tags are in silico extracted from the input WMS data (Fig. 2a) and aligned against a preconstructed unique 2b tag database (Fig. 2b). Here, the preconstructed unique 2b tag database is obtained as follows. We use a Type IIB enzyme (*CjepI*) to in silico digest all microbial genomes in GTDB[25] and Ensembl Fungi[26]. Then we compare the theoretically existent 2b tags for each species with all the others in the integrated microbial genome database to identify species-specific (or unique) 2b tags. Here theoretically existent 2b tags include all the 2b fragments generated by *CjepI* during the in silico restriction digestion. In total, there are 48,475 species in the preconstructed unique 2b-tag database. This first-round reads alignment against a massive number of reference genomes in database will generate preliminary profiling results (Fig. 2c). The first-round output also includes the genome coverage, taxonomic count, sequence count, and G-score of those identified species, which will be passed into a pretrained classifier (e.g., Random Forest) to

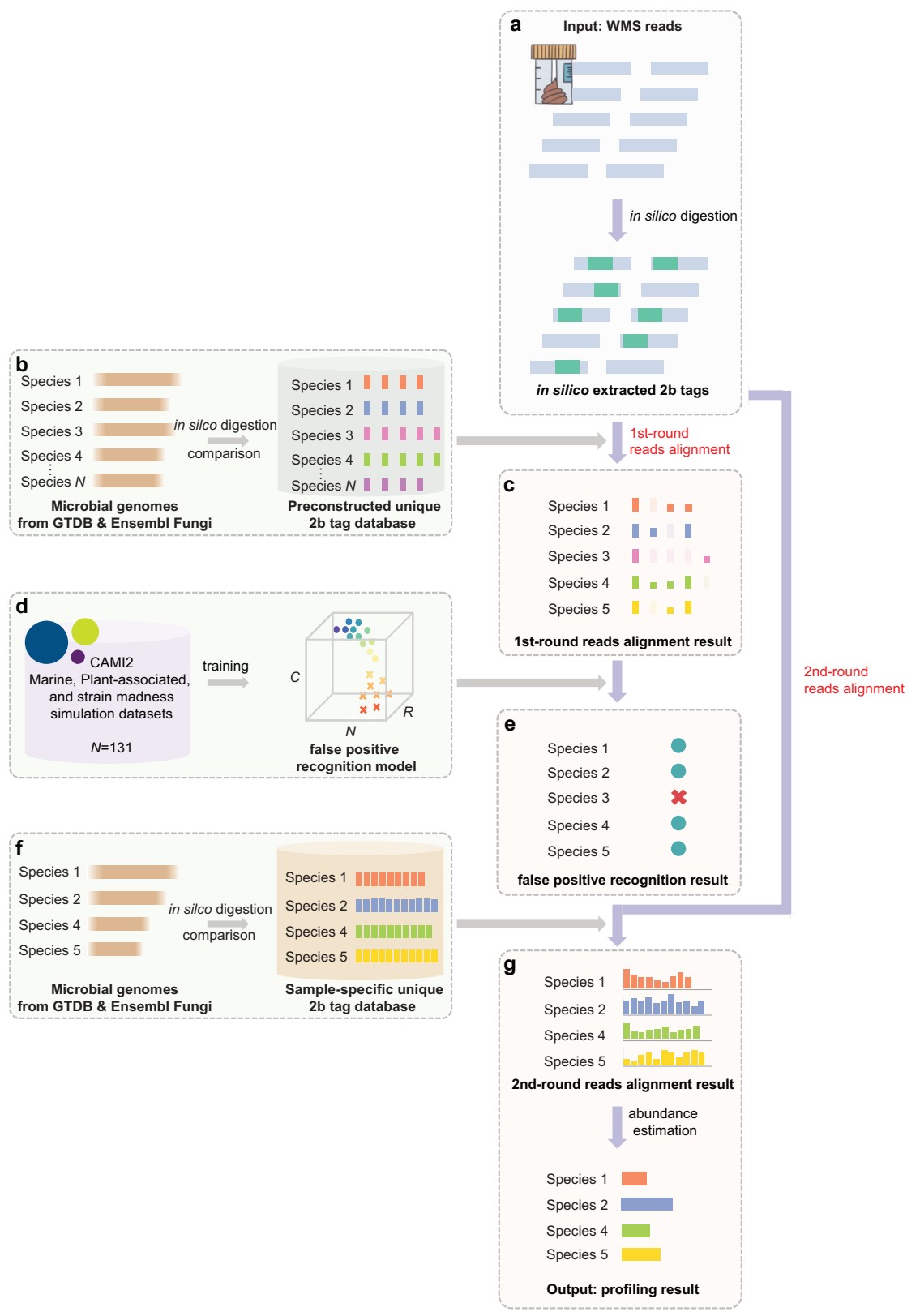

discriminate false positives from true positives (Fig. 2d, e). Theoretically existent 2b tags of those true-positive species determined by the machine-learning classifier will be compared to construct a sample-specific unique 2b-tag database (Fig. 2f). Due to the benefits from the drastic decrease in species number (typically on the order of hundreds for a microbiome sample) compared to that of the preconstructed database (i.e., 48,475),

the sample-specific unique 2b-tag database will contain much more (approximately twice) unique 2b tags for each species (Supplementary Fig. S2). Finally, the in silico extracted 2b tags from metagenomic data will be aligned to the sample-specific unique 2b tag database. This second-round reads alignment will generate final taxonomic profiling results (Fig. 2g). The two-round reads alignment strategy follows a reasonable logic flow of

**Fig. 2 | The workflow of MAP2B. a** For any input WMS data, 2b tags can be extracted by in silico digestion based on a given type IIB restriction enzyme. **b** Then, the WMS originated 2b tags will be mapped against a preconstructed unique 2b tag database. To construct the unique 2b tag database, we first in silico digested all microbial genomes in the GTDB and Ensembl Fungi, then compared the theoretically existed 2b tags from each species with all the other species within database. After comparison, species-specific 2b tags (unique 2b tags) were identified and gathered to construct a unique 2b tags database. **c** In the first round of reads alignment, we calculate the coverage, taxonomic count, sequence count and G score for each species. **d** Then the four features will be passed into a machine-learning model for false-positive recognition, which was preconstructed (trained) by CAMI2 datasets. **e** After first-round reads alignment and false-positive elimination by our machine-learning model, a high-precision species identification result will be generated. **f** Then, a sample-dependent unique 2b tag database will be constructed based on the species identification result, which aims to accurately estimate the taxonomic abundance of identified species by increasing their unique 2b tags. The procedure of reconstructing unique 2b database is similar to (**c**) but only uses the genomes from the identified species instead of 48,475 microbial species. **g** In the second round of reads alignment, we calculate the average sequencing coverage of all 2b tags for each species to finally estimate their taxonomic abundance (see "Methods").

qualitative analysis first and then quantitative analysis, which can generate a highly accurate profiling result.

## Benchmarking MAP2B with state-of-the-art metagenomic profilers

To evaluate the performance of MAP2B in decoding the taxonomic structure of microbiome samples using WMS data, we randomly selected microbial genomes in the NCBI RefSeq (v.24/06/2021) and systemically simulated a WMS dataset with sequencing depth varying from 7.5 to 150 million reads and species richness varying from 10 to 500 ("Methods"). Then we compared the profiling results generated by MetaPhlAn4, mOTUs3, Bracken, Kraken2, KrakenUniq, and MAP2B through a series of measurements (detailed information regarding the software, databases, and parameters utilized in the comparison can be found in the Supplementary Information). In particular, we used Precision, Recall, and F1 score to evaluate the species identification, and used L2 similarity (1 - L2 distance) and BC similarity (1 - Bray-Curtis dissimilarity) to evaluate the abundance estimation (Fig. 3). We found that: (1) In terms of species identification, regardless of the sequencing depth, although Bracken, Kraken2, and KrakenUniq have increasing identification performance in species richer samples, MetaPhlAn4 and mOTUs3 outperformed them in Precision (average Precision for MetaPhlAn4, mOTUs3 vs. Bracken, Kraken2 and KrakenUniq is 0.829, 0.444 vs. 0.052, 0.102 and 0.190), Recall (0.868, 0.489 vs. 0.333, 0.329, 0.303), and F1 score (0.846, 0.445 vs. 0.085, 0.145, 0.211), which is consistent with previous benchmarking work[13,14]; (2) As for the abundance estimation, the L2 (or BC similarity) revealed a performance rank (excluding MAP2B) as MetaPhlAn4 (with mean L2 similarity 0.916 and mean BC similarity 0.861), mOTUs3 (0.813, 0.474), Bracken (0.802, 0.321), Kraken2 (0.785, 0.306) and KrakenUniq (0.768, 0.283); (3) MAP2B outperformed all state-of-the-art metagenomic profilers in all measurements regardless of the species richness and sequencing depth (mean Precision = 0.989, Recall = 0.988, F1 score = 0.988, L2 similarity = 0.994, and BC similarity = 0.989).

As we know, the number of identifiable species largely depends on the reference databases used by different metagenomic profilers. To minimize the influence of database discrepancies on evaluating the performance (especially the Recall), we selected the microbial genomes largely shared among reference databases used by multiple metagenomic profilers to simulate the WMS data. After comparing the profiling results generated by the above profilers, we found that the potential bias introduced by different reference databases on measuring the Recall has been minimized, e.g., the Recall for all profilers reach up to 0.99 or 1 (Supplementary Fig. S3). Nevertheless, the conclusion that MAP2B exhibits a better performance in species identification when evaluated by Precision, Recall and F1 score, as well as its higher L2 similarity and BC similarity compared to others remains valid. For example, the performance based on Precision ranking is: MAP2B, 0.997; MataPhlAn4 0.967; mOTUs3, 0.931; Kraken2, 0.907; Bracken, 0.868; and KrakenUniq, 0.828, while the ranking based on L2 similarity is: MAP2B, 0.995; Bracken, 0.993; Kraken2, 0.988; KrakenUniq, 0.984; mOTUs3, 0.981 and MataPhlAn4, 0.972.

In addition, estimating accurate species abundance can be challenging when microbial genomes are not present in the reference database, which is a common issue for all metagenomic profilers (due to the reliance on reference databases). To demonstrate the limited influence of unknown species' influence on MAP2B's performance, we held out 1000, 5000, and 10,000 microbial genomes from the GTDB during the database construction and then simulated WMS sequencing data based on these held-out genomes and evaluated MAP2B's performance using these independent datasets (Supplementary Fig. S4). Our preliminary results showed no significant drop in the performance of species identification or abundance estimation (F1 = 0.922, L2 similarity = 0.954 when holding out 1000 to 10,000 independent genomes for WMS data simulation and testing). To further evaluate the robustness of MAP2B in handling complex scenarios such as mutations in sequencing data, we simulated WMS data with varying mutation rates of 1%, 2%, and 3%, which are representative of the nucleotide divergence observed between different strains within the same species (Supplementary Fig. S4). Our comparison results show that even with a high mutation rate of 3% in the sequencing data, MAP2B maintained a high level of precision (and accuracy), achieving an F1 score of 0.989 (and an L2 similarity of 0.990). These results suggest that MAP2B can effectively handle genomic variations in metagenomic data, making it a reliable tool for accurate taxonomic profiling of complex microbial communities, especially in eliminating false positives in species identification. Taken together, we illustrated a superior performance of MAP2B in both species identification and abundance estimation compared with other state-of-the-art metagenomic profilers based on in silico simulation metagenomes.

## The performance of MAP2B in mock WMS data

To further test the capability of MAP2B, we compared the profiling results of real WMS data (-10GB) of an ATCC mock sample (MSA 1002) generated by MAP2B and existing metagenomic profilers. MSA 1002 is a genomic DNA mixture of 20 microbial species with equal abundance (5% for each). It was prepared from fully sequenced, characterized, and authenticated ATCC Genuine Cultures that were selected based on relevant phenotypic and genotypic attributes (such as Gram stain, GC content, genome size, and spore formation) and has been widely used to distinguish incorrect classifications introduced by experimental and computational factors[13]. We found that MAP2B profiling results can better decode the taxonomic structure of the mock community, e.g., average F1 score (and L2 similarity) in species identification is 1 (0.923) by MAP2B compared to 0.950 (0.888), 0.930 (0.913), 0.471 (0.908), 0.571 (0.846), and 0.851 (0.840) by MetaPhlAn4, mOTUs3, Bracken, Kraken2, and KrakenUniq separately (Fig. 4). Notably, no false positive was reported for mock samples by MAP2B, while other metagenomic profilers reported 25% to 70% false-positive species in their profiling results.

## Application of MAP2B in real WMS data

To demonstrate the practical advantages of MAP2B, we applied MAP2B to a real WMS dataset collected from a human cohort study named PRISM[15], a study aimed at understanding gut microbiome structure

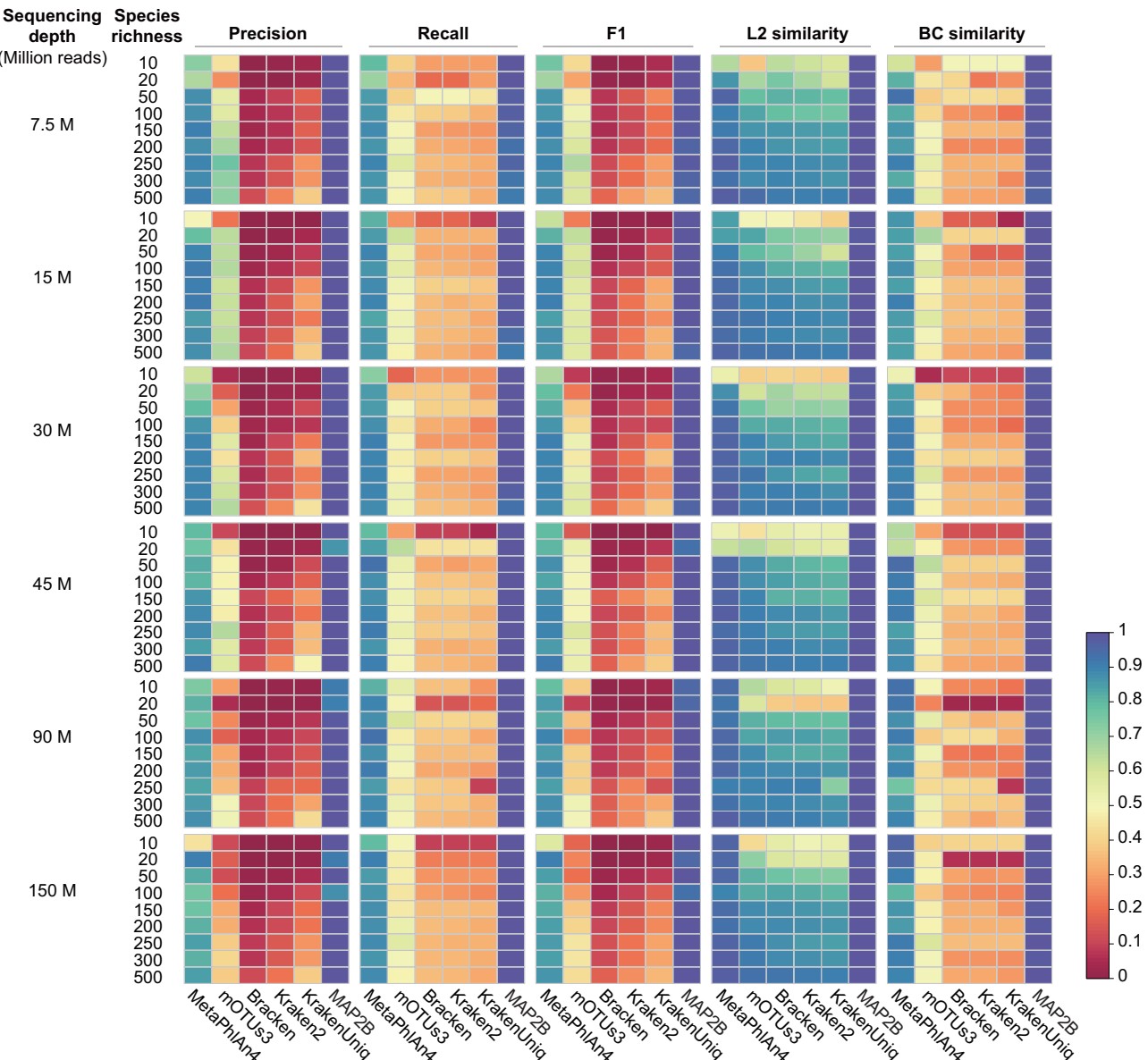

**Fig. 3 | Performance comparison of MAP2B with state-of-the-art metagenomic profilers in species identification and abundance estimation using a set of simulation metagenomes.** From left to right, the profiling results generated by different metagenomic profilers (such as MetaPhlAn4, mOTUs3, Bracken, Kraken2, KrakenUniq, and MAP2B) were compared with ground truth and illustrated by the precision, recall, F1 score, L2 similarity, and BC similarity. From top to bottom, the simulated sequencing depth increases from 7.5 M to 150 M, and the species richness increases from 10 to 500 under each sequencing depth.

and metabolic activity in inflammatory bowel disease (IBD) using metagenome sequencing data and metabolic profiles of individuals ($n = 220$) with and without IBD. We hypothesized that the accurate taxonomic features generated by MAP2B (such as the abundance profile $T_i$ and genome coverage profiles $C_i$) could be better associated with disease status and metabolic activity. To test this hypothesis, we first performed Principal Coordinates Analysis (PCoA) and permutational multivariate analysis of variance (PERMANOVA) to visualize and quantify the differences between IBD patients and healthy controls. We then employed MiMeNet[28] and mNODE[29] to predict the metabolomic profiles based on the taxonomic profiles. All the above analyses were conducted using the output of different metagenomic profilers and compared to demonstrate the superior performance of MAP2B.

To test if the taxonomic profiles (both the abundance and coverage profile) by MAP2B can better distinguish IBD from non-IBD, we compared the PCoA plots and PERMANOVA pseudo-F statistic based on taxonomic profiles generated by different metagenomic profilers

for the discovery cohort ($n = 155$) and validation cohort ($n = 65$) separately (Fig. 5a, "Methods"). We found that when the abundance profiles were used in PCoA and PERMANOVA, the community-level difference between IBD and non-IBD is similar regardless of the metagenomic profiler used, e.g., the F values are approximately 5.9 for all profilers in the discovery cohort. However, if we use the taxonomic coverage in PCoA and PERMANOVA, the difference between IBD and non-IBD is much more significant in the PCoA plot, and the F value reaches 9.3 for the discovery cohort. Moreover, we observed similar comparison results of PCoA and PERMANOVA on the validation cohort (Supplementary Fig. S5a): F value reaches 3.7 in MAP2B (using coverage) compared to approximately 2.6 in other profilers (using abundance). This suggests a bottleneck in discriminating disease status using taxonomic abundance, while taxonomic coverage can better distinguish IBD from non-IBD.

We then sought to test if the more accurate taxonomic profiles produced by MAP2B can better predict metabolic activities. Specifically,

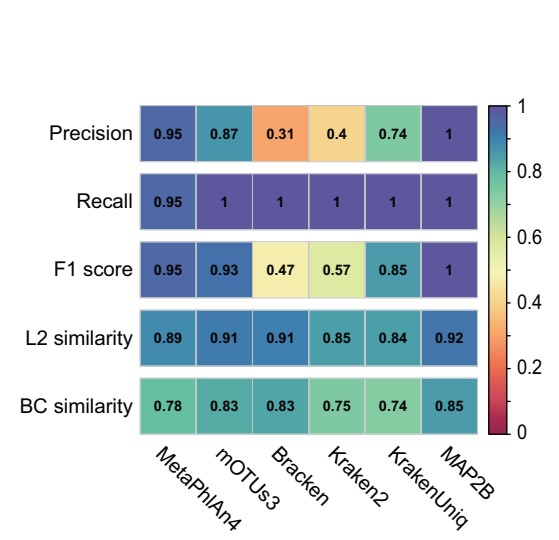

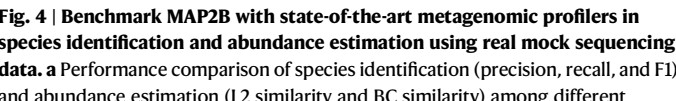

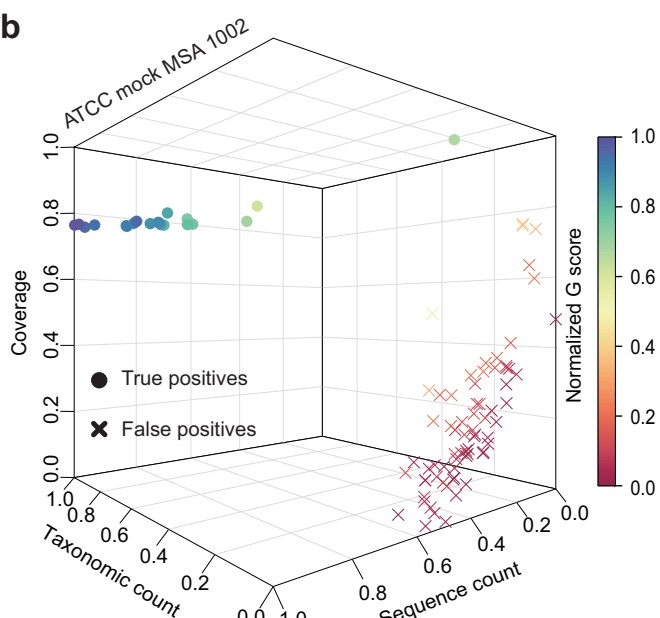

**Fig. 4 | Benchmark MAP2B with state-of-the-art metagenomic profilers in species identification and abundance estimation using real mock sequencing data. a** Performance comparison of species identification (precision, recall, and F1) and abundance estimation (L2 similarity and BC similarity) among different metagenomic profilers on the real sequencing data of ATCC mock MSA 1002. **b** 3D scatter plot shows the distributions of true-positive and false-positive species identified by MAP2B. A clear boundary between the 20 true positives and false positives was observed, which is in line with the ground truth.

we employed MiMeNet[28] and mNODE[29] to predict metabolomic profiles based on the taxonomic profiles. We first performed fivefold cross-validation in the discovery cohort ($n = 155$ paired microbiome-metabolome samples, "Methods") to determine the best hyperparameter set and then predicted metabolite concentrations for the validation cohort ($n = 65$ paired microbiome-metabolome samples, "Methods"). To compare the prediction performance, we measured the SCC (Spearman Correlation Coefficient) of a metabolite between its true concentration values and the predicted values by microbial composition across all samples in the validation cohort. We adopted the same prediction procedure for taxonomic profiles from different metagenomic profilers and intended to find out which metagenomic profiler gives the most useful taxonomic profile for such a prediction task. (1) The taxonomic abundance obtained by MAP2B outperformed others in the mean SCC computed by averaging SCCs of all metabolites (Fig. 5b), e.g., MAP2B (0.337 by MiMeNet) was ranked the first and followed by mOTUs3 (0.329). (2) We observed the largest number of metabolites with SCCs larger than 0.5 by using MAP2B's taxonomic abundance among all methods used for taxonomic profiling (Fig. 5c). We identified 119 accurate predictions with MiMeNet using MAP2B profiles, while only 108 accurate predictions were identified based on MetaPhlAn4 profiles, ranked the second. (3) MAP2B used the lowest taxonomic features to achieve this high prediction performance (Fig. 5d). Only 238 microbial species predicted by MAP2B are used in the metabolome prediction, while mOTUs3 and MetaPhlAn4 need 461 and 367 features. (4) The above observations by mNODE are quite in line with MiMeNet (Supplementary Fig. S5b, c). Taken together, we demonstrated that the machine-learning-driven accurate taxonomic profiling provided by MAP2B can best discriminate IBD from non-IBD, and the taxonomic abundance and coverage by MAP2B can more accurately predict metabolomic profiles with the fewest features compared to state-of-the-art metagenomic profilers.

## Discussion

Currently, existing reference-based metagenomic profilers can be divided into three categories based on their algorithms and output abundance type: (1) DNA-to-DNA methods such as Bracken[16], Kraken[21], PathSeq[30], and Clark[31], which use whole microbial genomes as the reference and usually output sequence abundances in their profiling results; (2) DNA-to-Protein methods such as Kaiju[32] and Diamond[33] which used whole protein sequence as the reference and usually output sequence abundance; (3) DNA-to-Marker such as MetaPhlAn and mOTUs which use universal single-copy markers as the reference and output taxonomic abundance[13]. All three types of methods suffer from false-positive and false-negative issues in species identification[27].

There are four major reasons for low Precision, i.e., false-positive identifications especially by DNA-to-DNA methods: (1) Conserved and low complexity regions in microbial genomes will lead to multi-alignment for the sequencing reads and then generate misclassifications[22]; (2) About 1–5% of human reads are highly similar to microbial genomes, it will further confuse the microbial species identification[13], and it is not practical to remove host DNA by current experimental approaches[34]; (3) It is impossible to distinguish false positives from true positives using relative abundance as the threshold, because as we have illustrated false positives are not necessarily low abundant taxa[24], and previous studies reported that less than 0.1% of the DNA may derive from microbes of interest[35, 36]; (4) Additional spurious identifications can also result from contamination in the reference genome databases themselves[37].

On the other hand, DNA-to-Marker methods also suffer from low Recall more than DNA-to-DNA methods, i.e., false-negative identifications in microbial profiling[13], because DNA-to-Marker methods such as MetaPhlAn and mOTUs have less identifiable species in their reference databases compared to DNA-to-DNA methods, which is caused by (1) missing of universal markers in some microbial genomes; (2) incomplete genome information in publicly available databases which may contribute to the missing marker issue, and (3) unfriendly reference database customization[24]. Notably, it is possible that the markers of low abundance species may not be fully detected in the sequencing data, especially if the markers do not cover the entire genome of the microbe[38].

Considering the biological relevance, the development of metagenomic profilers that are able to provide taxonomic abundance (instead of sequence abundance) is highly encouraged[27]. MAP2B does

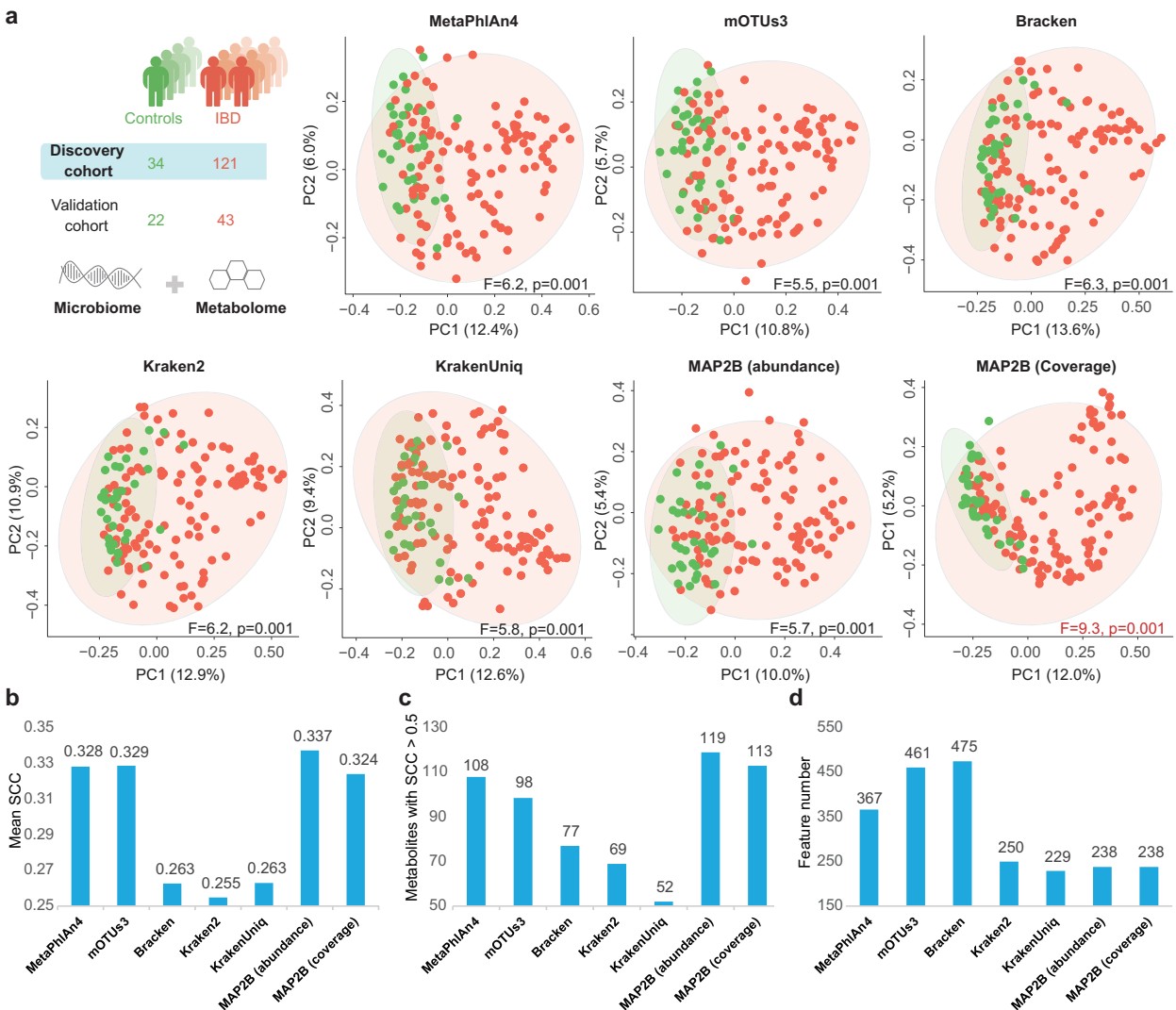

**Fig. 5 | Assessing MAP2B's microbial profiles in disease discrimination (discovery cohort) and prediction of metabolomic profiles (using MiMeNet).**
**a** PCoA plots for the discovery cohort ($n = 155$) based on the taxonomic profiles generated by different profilers. The ellipses with 95% CI are drawn to illustrate the difference between IBD (red dots) and non-IBD (green dots) in PCoA. F values and *P*-values of the PERMANOVA test are also marked on the bottom of each plot to quantify the difference in disease status. **b** Comparison of prediction results by using different taxonomic profiling via mean SCC of the metabolite between its true values and predicted values across all individuals in the validation cohort. **c** Comparison of the number of metabolites with SCCs larger than 0.5 among different taxonomic profiling results. **d** Comparison of the number of taxonomic features used by different metagenomic profilers in the prediction for metabolomic profiles. The prediction results in (**b**) and (**c**) were generated by the MiMeNet.

not belong to any of the three existing categories yet can produce taxonomic abundance. While a typical DNA-to-Marker method often uses relatively long universal single-copy markers, we leveraged thousands of short species-specific 2b tags that are distributed all over the microbial genome for species identification. The taxonomic markers for conventional DNA-to-Marker methods are often located at a particular region of a genome. In WMS data, we often can't have a complete alignment of the full-length marker genes and thus produce a low recall of microbial identification. Differently, the short 2b tags in MAP2B are usually distributed far apart across a microbial genome, and we often observe a relatively high sequence coverage on each taxonomic marker in practice, significantly mitigating the low-recall issue in the conventional DNA-to-Marker methods. Notably, MAP2B combines merits from both DNA-to-Marker and DNA-to-DNA methods to perform species identification and abundance estimation. For example, MAP2B preconstructed a marker reference database similar to what a typical DNA-to-Marker method did. Given these unique 2b tags are located far apart from each other in the microbial genomes

and can be recognized as the reduced genomes, the alignment of massive reads to the reference reduced genome database for taxonomic profiling is highly consistent with what conventional DNA-to-DNA methods did. These combined characteristics from DNA-to-Marker and DNA-to-DNA methods enable MAP2B to provide comprehensive genetic features such as species' genome coverage, taxonomic count, and sequence count at the same time, laying a solid foundation for its excellent performance in eliminating false positives.

MAP2B was motivated and inspired by 2bRAD-M computational pipeline for processing 2bRAD sequencing data. Notably, All the unique 2b tags mentioned here can be enriched and sequenced from any microbiome samples with 2bRAD-M protocol we developed previously. With this protocol, we generated a novel 2bRAD metagenomic data type, which is distinct from either conventional 16S rRNA sequencing data or metagenomics data. Therefore, the false-positive reduction algorithm in the MAP2B also offers a noteworthy opportunity to improve the taxonomic profiling accuracy of 2bRAD-M[24] based on 2bRAD sequencing data. Other than adding a machine-learning

model to correct high-positive issue to 2bRAD-M, we also updated the reference database, including 48,475 identifiable microbial species, as well as the adjustment in input data types, which now include both conventional metagenomic data and 2bRAD sequencing data. These improvements enable MAP2B to have versatile potential to perform microbiome analysis from the massive publicly available metagenomic sequencing datasets.

Species identification issues such as false positives and false negatives are the global challenges that faced by all existing metagenomic profilers. To our knowledge, there have been many efforts by state-of-the-art metagenomic profilers to deal with these issues. For example, as the main cause for false-positive identifications, the ambiguous reads (in reads alignment) are: (1) spited evenly across all matches in Kraken[21]; (2) pushed count to the lowest common ancestor of matched genomes in Bracken[16]; (3) in synergy with unique *k-mers* for abundance estimation[22]. However, optimization around a single taxonomic feature (abundance) cannot effectively solve the false-positive issue. Recently, the coverage of genomes has been proved useful in determining the existence of species[39], while we believe the key to addressing the species identification issue is involving more biologically significant taxonomic features (e.g., the taxonomic features of coverage, taxonomic count, sequence count, and G score in this study) in false-positive recognition and expanding the identifiable species as large as possible.

MAP2B does have some limitations. For instance, it has classification issues for unknown species for which we have no genome reference. (Note that all other reference-based metagenomic profilers suffer from this issue.) To alleviate this problem, we employed GTDB (v.202) and Ensembl Fungi as the reference database, which includes more than 48,000 microbial species in total. Additionally, any new version database with a higher number of identifiable species can be accepted by MAP2B. However, GTDB has more than 40% inconsistent taxonomic annotation with NCBI RefSeq since GTDB is based on ANI of microbial genomes instead of morphological classification. In order to mitigate the impact of annotation differences on the interpretation of microbial data, we provide both RefSeq and GTDB versions of microbial databases in our GitHub repository. Moreover, the increasing number of Metagenomic-Assembled Genomes (MAGs) have been identified in microbiome studies focusing on a range of habitats (such as human, gut, oral, soil, marine, etc.), which have largely expanded our capability to explore the unknown microbial world. In our next work, for a microbiome taxonomic profiling task focusing on the specific habitats (such as gut microbiota) or disease status, we can establish a habitat/disease-centric reduced reference genome database with both existing reference genomes and high-quality MAGs to address the challenges. We believe that MAP2B will serve as a strong candidate metagenomic profiler for decoding the taxonomic structure, eliminating false positives in species identification, and consequently enhancing the interpretation of metagenomic data in microbiome studies.

## Methods
### Rationale of avoiding false-positive and improving false-negative identifications
By comparing 2b fragments (or tags) generated by in silico digestion of all publicly available microbial genomes in GTDB, we found there are some special 2b tags contained in each species that have no duplications in any other species, which can serve as markers for species identification and abundance estimation. We named these special 2b tags as species-specific 2b tags or unique 2b tags. In the database combined with GTDB and Ensembl Fungi, we found that there are, on average, 8607 unique 2b tags for each species (digested by *CjepI*), presenting the preconstructed unique 2b tag database for MAP2B. Due to the special marker selection, MAP2B can naturally avoid confusion from conserved and low complexity regions in microbial genomes[24].

Notably, we have proved that a single type IIB enzyme, e.g., *BcgI*, would meet the requirement for decoding microbial samples with high accuracy while the combination of different IIB enzymes will be marginal in improving its accuracy[24].

On the other hand, MAP2B significantly improves the Recall in its profiling results compared to other profilers by: (1) expanding the identifiable bacterial and archaeal species to 47,894 (based on the GTDB database release 202[25]) and 581 fungal species (based on the Ensemble release 48[26]). To our knowledge, MAP2B has the largest number of identifiable species compared to existing DNA-to-Marker metagenomic profilers, e.g., mOTUs3 (~33,000) and MetaPhlAn4 (~24,000); (2) the widely distributed unique 2b tags across the microbial genome. For example, when determining a true positive, reads from existent microbes should distribute relatively uniformly across the genome rather than being concentrated in one or a few locations[22]. Therefore, the markers that are widely distributed across the microbial genome will provide more precise microbial identification. In previous studies, we have proved that unique 2b tags (regardless of which enzyme is used) are widely and evenly distributed on microbial genomes[24].

### Difference between MAP2B and Marker-based metagenomic profilers
It is worth noting that the database of MAP2B differs fundamentally from the databases of traditional universal marker methods such as MetaPhlAn. This is because MAP2B does not rely on representative sequences for each species. Instead, MAP2B marks species-specific 2b tags for each genome in its database (Supplementary Fig. S6). Unlike traditional methods that rely on universal markers, the selection of taxa-specific 2b tags can be conducted separately at any taxonomy level, as we can always compare the 2b tags of one specific genome with those of all other genomes from different species (or any higher taxonomy levels) to obtain species-specific (or other taxonomy level-specific) 2b tags, without considering the markers' similarity within the same species as in the case of traditional universal markers methods. Indeed, strain/species/genus/family/order/class/phylum-specific 2b tags can be generated separately, and the selection process is not affected by the variation among different genomes (e.g., conspecific strains) within the same taxa. This means that the computation of species-specific 2b tags is independent of the set of strain-specific 2b tags. Therefore, in the construction of the species-specific 2b tag database that contains numerous conspecific strains, we simply record all species-specific 2b tags for each of the 259,388 genomes in our database.

### Calculation of the four features
Based on the unique nature of species-specific 2b tags, we proposed four features, including genome coverage, taxonomic count, sequence count, and G-score (Supplementary Fig. S7), which can be calculated from the first-round reads alignment in MAP2B (Fig. 2a–c).

Consider a database of microbial genomes (e.g., GTDB[25]). For species-$i$ in this database, we denote its total number of 2b tags generated by in silico digestion of its genome as $H_i$. Among the $H_i$ tags, there are $E_i$ tags that are single-copy within species-$i$'s genome and are unique to species-$i$ w.r.t all other species in the database. Given an input WMS dataset, we in silico extract 2b tags, map them to the species-specific 2b tags and denote the number of tags unique to species-$i$ as $Q_i$. Among the $Q_i$ tags unique to species-$i$, there are $U_i$ distinct or nonredundant ones.

The genome coverage of species-$i$, denoted as $C_i$, is defined as:

$$C_i = \frac{U_i}{E_i}, \tag{1}$$

which quantifies the percentage of unique 2b tags present in the sequencing data.

In the WMS data of complex microbial communities, typically, we have the genome coverage $C_i < 1$, because some of the species-specific or unique 2b tags (e.g., the two tags highlighted in red dashed boxes) are absent in the sequencing data. To take this into account, we infer the actual number of sequenced unique 2b tags, denoted as $\widetilde{Q}_i$, by the genome coverage correction, i.e.,

$$\widetilde{Q}_i = \frac{Q_i}{C_i}. \tag{2}$$

The taxonomic count of species-$i$, denoted as $N_i$, is simply calculated as the average count of the sequenced unique 2b tags, i.e.,

$$N_i = \frac{\widetilde{Q}_i}{E_i} = \frac{Q_i}{U_i}. \tag{3}$$

The sequence count of species-$i$, denoted as $R_i$, is defied to be the average count of the inferred unique 2b tags sequenced per read:

$$R_i = \frac{\widetilde{Q}_i}{R} = \frac{Q_i E_i}{U_i R}, \tag{4}$$

here $R$ is the total number of reads in the WMS data, which might vary a lot across different samples.

G score ($G_i$) is simply the geometric mean of $U_i$ and $Q_i$, i.e.,

$$G_i = \sqrt{Q_i * U_i}. \tag{5}$$

The four features above are then log-transformed before inputting into the false-positive recognition model.

## MAP2B workflow
First, we downloaded 258,406 bacterial and archaeal genomes from the GTDB release 202[25] and 982 fungal genomes from the Ensemble release 48[26]. We in silico digested all the microbial genomes using _CjepI_ as the type IIB enzyme. Comparing theoretically existent 2b tags across different species, we found an average of 8,607 unique tags for each species. This allows us to construct a unique 2b tag database that contains 2b tags unique to each of 48,475 (47,894 + 581) microbial species in the GTDB and Ensemble.

Secondly, in silico digestion also works for WMS data, generating 2b tags that can be mapped against the preconstructed unique 2b tag database for species identification. In the first round of reads alignment, we calculate the coverage, taxonomic count, and sequence count for feeding the machine-learning model to recognize false positives (which is trained using CAMI2 simulation data). After generating species identification results, a sample-specific unique 2b tag database will be constructed, aiming to accurately estimate the taxonomic abundance of identified species by increasing their unique 2b tags.

The taxonomic abundance ($T_i$) of a given species can be calculated as the ratio between cells of a species and all cells in the microbial community. By calculating the average coverage of all theoretically existent 2b tags ($H_i$) for each species, we are able to estimate the number of cells belonging to a species present in a sample at a given sequencing depth. In the second round reads alignment, due to the increased unique 2b tags in the sample-specific unique 2b tag database, we estimate the relative abundance of each microbial species using an adjusted formula as below:

$$T_i = \frac{Q_i/H_i}{\sum_{j=1}^{n} Q_j/H_j} \tag{5}$$

The taxonomic coverage $O_i$ is similar to $C_i$ calculated by Eq. (1); the only difference between $C_i$ and $O_i$ is the unique 2b tag databases used for reads alignment, e.g., $C_i$ is generated in the first round of reads alignment by searching against the preconstructed unique 2b tag database while $O_i$ is generated in the second round of reads alignment by searching against the sample-specific unique 2b tag database.

$$O_i = \frac{U_i}{E_i}, \tag{6}$$

## Benchmarking MAP2B
To evaluate the performance of MAP2B, we simulated a series of simulation data. Specifically, for Fig. 3, we first simulated microbial profiles ($n = 54$) varied in species richness (from 10 to 500) with known taxonomic abundance (taxonomic abundance was created randomly from a log-normal distribution using the function rlnorm in the R language with the following parameters: meanlog = 0 and sdlog = 1). Then, for each species richness, we generated different reads (from 7.5 million to 150 million) using the Wgsim (https://github.com/lh3/wgsim, with default parameters) to simulate changeable sequencing depth in the real world. In order to ensure the randomness and generality of the benchmarking simulation data, source genomes were selected from the intersection of RefSeq and GTDB. The simulation scripts for WMS data can be found at https://github.com/sunzhengCDNM/MAP2B. Notably, we generated both sequence abundance and taxonomic abundance as ground truth (e.g., for a given taxonomic abundance, its sequence abundance can be inferred accordingly: taxonomic abundance equals sequence abundance divided by their genome length), the former is used to benchmark DNA-to-DNA metagenomic profilers (e.g., Bracken, Kraken2, and KrakenUniq) while the latter is used to benchmark DNA-to-Marker methods (e.g., mOTUs3 and MetaPhlAn4) and MAP2B.

To minimize the influence of different reference databases on measuring the recall, we further selected the shared microbial genomes between different metagenomic profilers (e.g., mOTU3, MetaPhlAn4, and Kraken2) as source genomes for simulating the WMS data. Since selecting the intersection of different metagenomic profilers' reference genomes dramatically decreased the number of source genomes for simulation, we slightly adjusted the species number (from 25 to 400) and sequencing depth (from 8 million to 102 million) in the simulation data ($n = 30$) for Supplementary Fig. S3.

We conducted additional simulations ($n = 27$) to further evaluate the performance of MAP2B on WMS data using independent microbial genomes from GTDB. We simulated datasets with 1000, 5000, and 10,000 genomes from GTDB that were held out during the construction of the unique 2b tag database. These genomes were then used as the source to generate simulated data with varying sequencing depth and species richness. To evaluate the effect of genomic variations, we also simulated datasets ($n = 27$) with different mutation rates of 1, 2, and 3%. We used the Wgsim software and set the "-r" parameter to 0.01, 0.02, and 0.03, respectively, to control the mutation rates. The results of these simulations are presented in Supplementary Fig. S4. As for the usages of state-of-the-art metagenomic profilers, the default parameters were employed as previously described[24]; please see the Supplementary Information for more details. To ensure the accuracy of benchmarking, we used the taxid (NCBI RefSeq) when processing the comparisons between ground truth and profiling results by different metagenomic profilers.

## Principal coordinates analysis and PERMANOVA test
Parallel-Meta 3.5[40] was used to draw the PCoA plots, which are based on Bray-Curtis (BC) dissimilarity matrixes derived from the taxonomic profiles by different metagenomic profilers. Differences in beta-diversity (and PCoA) based on BC dissimilarity were determined using permutational multivariate analysis of variance (PERMANOVA)

with 999 random permutations. The test statistic is a pseudo-F ratio, similar to the F-ratio in ANOVA. It compares the total sum of squared dissimilarities (or ranked dissimilarities) among objects belonging to different groups to that of objects belonging to the same group. Larger F-ratios indicate more pronounced group separation.

## Metabolomic profiles prediction by MiMeNet and mNODE

To explore whether the microbial composition inferred from MAP2B is most helpful in capturing microbial activities, we compared the accuracy of predicting metabolomic profiles based on the taxonomic profiles by different metagenomic profilers. Specifically, we performed fivefold cross-validations in the discovery cohort of PRISM (individual = 155) to determine the best hyperparameter set and then generated predictions for metabolite concentrations based on the taxonomic profiles in the validation cohort of NLIBD (individual = 65).

MiMeNet (Microbiome-Metabolome Network)[28] and mNODE (Metabolomic profile predictor using Neural Ordinary Differential Equations)[29] are computational methods used in this study to predict metabolomic profiles based on microbial compositions and later integrate microbiome and metabolome data to uncover microbe-metabolite interactions in a data-driven manner. MiMeNet uses neural networks (i.e., multilayer perception) to predict metabolite abundances from microbe features, and mNODE is based on a state-of-the-art family of deep neural network models (i.e., neural ordinary differential equations). For both methods and taxonomic profiles from all metagenomic profilers, only microbial taxa with a prevalence larger than 10% are kept. The software is made freely available at https://github.com/YDaiLab/MiMeNet and https://github.com/wt1005203/mNODE.

## Statistics and reproducibility

In evaluating overall performance, we utilized precision, recall, and the F1 score to assess the accuracy of species identification. Precision represents the ratio of true-positive species to the total species identified by a method. Recall is the ratio of true-positive species to the total species present in a sample. The F1 score is the harmonic mean of precision and recall. As for abundance estimation accuracy, we employed L2 similarity (1 - L2 distance) and Bray-Curtis similarity (1 - BC dissimilarity). Scripts used for generating simulation data for reproducibility purposes are available on our GitHub repository in the folder "Manuscript/Figure3/WMS simulation". Our sample size references benchmark works like the CAMI2 simulation datasets (minimum 10 for different scenarios). Accordingly, we generated 54, 30, and another 54 simulation datasets with varied sequencing depth and species richness to compare the performance of different metagenomic profilers using random NCBI genomes (Fig. 4), shared genomes (Supplementary Fig. S3), and unknown or highly mutated genomes (Supplementary Fig. S4). All the microbial species and genomes used for generating the simulation data were randomly selected. No data were excluded from the analyses, and the investigators were not blinded to allocation during experiments and outcome assessment.

## Reporting summary

Further information on research design is available in the Nature Portfolio Reporting Summary linked to this article.

## Data availability

The WMS data of the ATCC MOCK MSA 1002 generated in this study have been deposited in the NCBI SRA database (and Figshare) under PRJNA1006621 (or can be downloaded from https://doi.org/10.6084/m9.figshare.21627077.v3). The computational pipeline of 2bRAD-M is licensed under the MIT license. The MAP2B computational pipeline and related database files are publicly available at GitHub (https://github.com/sunzhengCDNM/MAP2B).

## Code availability

All source data and codes for the generation of figures and tables in the manuscript can be accessed at GitHub (https://github.com/sunzhengCDNM/MAP2B/tree/master/Manuscript) or Zenodo (https://zenodo.org/record/8265883).

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

## Acknowledgements
This work was supported by the National Institutes of Health grant number R01AI141529 (Y.Y.L.), UH3OD023268 (S.T.W.), K99HL163519 (Z.S.), and the Charles A. King Trust Postdoctoral Fellowship (Z.S.).

## Author contributions
Z.S. and Y.Y.L. designed the project. Z.S. and J.L. developed the MAP2B code. M.Z. simulated the WMS data. Z.S. analyzed all the data with assistance from M.Z., J.L. and T.W. Z.S., T.W., S.H., S.W. and Y.Y.L. interpreted the results. Z.S. and Y.Y.L. prepared the manuscript. Z.S., M.Z., T.W., S.H. and S.W. edited and reviewed the manuscript. All authors approved the manuscript. Y.Y.L. supervised the study.

## Competing interests
The authors declare no competing interests.
