## [Peer Review File · Nature Communications]

REVIEWERS' COMMENTS

Reviewer #3 (Remarks to the Author):

I was invited to assess whether the previous round of revision has reasonably addressed the concerns raised by Reviewer #1.

Reviewer #1 raised several concerns, all of which focused on the comparison between MAP2B vs. other profilers against the CAMI 2 challenge dataset. This reviewer complained that the comparisons were unfair, for the reasons that (1) MAP2B was trained on additional data (e.g., source genomes, GTDB) not available to other profilers during the CAMI 2 challenge; (2) the creation of unique evaluation metrics by the authors, which were not used by other profilers during the CAMI 2 challenge.

Personally, I found these comments somewhat puzzling. The comparison should be fair as long as the authors used the most up-to-date software version and genome database for all other profilers. It appears this is the case for the current study (e.g., line 98, page 3). Clearly, the authors did not extract performance data from the CAMI 2 challenge paper and redrew in their study. Furthermore, the CAMI 2 challenge dataset was only used in the first paragraph of the Results section to highlight the issue of false-positives, rather than for direct benchmarking against other profilers. To enhance clarity, the authors may consider summarizing the version numbers for both software and database used in their comparisons.

For the same reasons I mentioned above, the second concern raised by Reviewer #1 appears to be resolved. Additionally, the metrics employed for evaluation were indeed similar between this study and the CAMI 2 challenge. For example, the CAMI 2 challenge used unfrac distance while the authors used bray-curtis dissimilarity.

Overall, the authors have provided a well-structured and compelling rebuttal letter. However, I must admit that I encountered some difficulty in grasping the precise goal of this paper. The initial

illustration of the problem of false-positives aligns perfectly with the stated objective of developing a new profiler to eliminate false positives in metagenomic profiling. Nevertheless, when benchmarking MAP2B against other profilers, the evaluation was primarily focused on species identification and taxonomic abundance. This appears somewhat unfocused. Similar issues remain in other benchmarks, such as ATCC mock data and human metagenomic data. Then what did the authors want to claim here? Are you claiming a profiler that is "better in terms of eliminating false-positives and comparable to other profilers in terms of other metrics", or a profiler that demonstrates superiority in all ways? Obviously, claiming a superior approach in all respects would necessitate additional benchmarking and research efforts.

In summary, the authors have effectively addressed Reviewer #1's concerns, and the manuscript holds great promise. However, to strengthen the contribution of the paper, I recommend refining the focus and clearly articulating the primary claim, ensuring it aligns with the stated goal of developing a new profiler to eliminate false positives.

Reviewer #4 (Remarks to the Author):

I've reviewed the authors' response to the original Reviewer #2's comments at the request of the handling editor. I did not write the original review but I find that the authors have appropriately addressed the concerns raised by Reviewer #2 by adding the requested benchmarking results and clarifications.

Responses to Reviewer #3

I was invited to assess whether the previous round of revision has reasonably addressed the concerns raised by Reviewer #1.

Response: We are grateful to Reviewer #3 for taking the time to thoroughly review our revised manuscript. We sincerely appreciate the constructive and positive feedback provided, which underscores the significance and impact of our work. Next, we address each of the reviewer comments in order.

Reviewer #1 raised several concerns, all of which focused on the comparison between MAP2B vs. other profilers against the CAMI 2 challenge dataset. This reviewer complained that the comparisons were unfair, for the reasons that (1) MAP2B was trained on additional data (e.g., source genomes, GTDB) not available to other profilers during the CAMI 2 challenge; (2) the creation of unique evaluation metrics by the authors, which were not used by other profilers during the CAMI 2 challenge.

Personally, I found these comments somewhat puzzling. The comparison should be fair as long as the authors used the most up-to-date software version and genome database for all other profilers. It appears this is the case for the current study (e.g., line 98, page 3). Clearly, the authors did not extract performance data from the CAMI 2 challenge paper and redrew in their study. Furthermore, the CAMI 2 challenge dataset was only used in the first paragraph of the Results section to highlight the issue of false-positives, rather than for direct benchmarking against other profilers. To enhance clarity, the authors may consider summarizing the version numbers for both software and database used in their comparisons.

Response: We appreciate this excellent suggestion. In the previous version of our manuscript, we documented the version numbers for both tools and databases used in our comparisons within the methods section of our **Supplementary Information**. As advised, to enhance clarity, we have supplemented a statement in the main text where benchmarking is first mentioned as follows: "*Detailed information regarding the software, databases, and parameters utilized in the comparison can be found in the **Supplementary Information**.*"

For the same reasons I mentioned above, the second concern raised by Reviewer #1 appears to be resolved. Additionally, the metrics employed for evaluation were indeed similar between this study and the CAMI 2 challenge. For example, the CAMI 2 challenge used unifrac distance while the authors used bray-curtis dissimilarity.

Overall, the authors have provided a well-structured and compelling rebuttal letter. However, I must admit that I encountered some difficulty in grasping the precise goal of this paper. The initial illustration of the problem of false-positives aligns perfectly with the stated objective of developing a new profiler to eliminate false positives in metagenomic profiling. Nevertheless, when benchmarking MAP2B against other profilers, the evaluation was primarily focused on species identification and taxonomic abundance. This appears somewhat unfocused. Similar

issues remain in other benchmarks, such as ATCC mock data and human metagenomic data. Then what did the authors want to claim here? Are you claiming a profiler that is "better in terms of eliminating false-positives and comparable to other profilers in terms of other metrics", or a profiler that demonstrates superiority in all ways? Obviously, claiming a superior approach in all respects would necessitate additional benchmarking and research efforts.

Response: Based on this constructive feedback, we have made numerous revisions in the Results section, particularly in parts related to benchmarking, to more strongly emphasize the performance comparison in species identification. Furthermore, to ensure clarity regarding our paper's objectives and to aid readers in understanding our central aim, we have enhanced statements on this topic in several areas of the Abstract, Introduction, and Discussion sections. Please refer to our revised manuscript for detailed changes.

In summary, the authors have effectively addressed Reviewer #1's concerns, and the manuscript holds great promise. However, to strengthen the contribution of the paper, I recommend refining the focus and clearly articulating the primary claim, ensuring it aligns with the stated goal of developing a new profiler to eliminate false positives.

Response: We once again express our gratitude to Reviewer #3 for his/her thorough review of our manuscript. The insightful and constructive comments have greatly enhanced the quality of our manuscript. We hope our responses above have addressed her/his comments satisfactorily.

Responses to Reviewer #4

I've reviewed the authors' response to the original Reviewer #2's comments at the request of the handling editor. I did not write the original review but I find that the authors have appropriately addressed the concerns raised by Reviewer #2 by adding the requested benchmarking results and clarifications.

Response: We appreciate the time and effort Reviewer #4 has taken to review our responses to the comments raised by Reviewer #2. We are pleased to hear that our efforts to address the concerns, along with our addition of the requested benchmarking results and clarifications, have been deemed satisfactory.